# Predator-Induced Nocturnal Benthic Emergence: Field and Experimental Evidence for an Unknown Behavioral Escape Mechanism along the Coral Reef–Seagrass Interface

**Derrick C. Blackmon** [1,2] **and John F. Valentine** [1,2,*]

1   Dauphin Island Sea Lab, 101 Bienville Boulevard, Dauphin Island, AL 36528, USA
2   Department of Marine Science, University of South Alabama, Mobile, AL 36688, USA
*   Correspondence: jvalentine@disl.edu; Tel.: +1-251-861-2141 (ext. 2261)

**Abstract:** Previously, using plankton tows, and emergence and settlement traps, we documented persistent widespread nocturnal emergence, and planktonic redistribution, of benthic macroinvertebrates along the coral reef–seagrass interface at two geographically separated locations. We also documented that emergence intensity varies with distance from the reef, leading us to hypothesize that the spatial pattern of emergence is determined by the foraging patterns of nocturnally active, bottom-feeding, mid-level consumers (mainly grunts). In this second study, we coupled those previously published data with nocturnal fish surveys concurrently conducted along belt transects placed at the same locations as the emergence trap collections, and a controlled laboratory experiment, to test this hypothesis. The results of these analyses find that variability in the density of nocturnally active, bottom-feeding fish is strongly positively correlated with emergence intensity, regardless of site or season. Results from the laboratory experiments show that nocturnal invertebrate emergence is significantly higher in the presence of one bottom-feeding fish (the blue-striped grunt *Haemulon sciurus*) than in microcosms that do not contain this predator. Overall, this study shows that such processes may explain how benthic prey can avoid capture by nocturnally active, bottom-feeding predators and persist in the predator-rich seagrass habitats that surround coral reefs in the Florida Keys National Marine Sanctuary. This study also points out the need to consider nocturnal processes when studying seagrass biodiversity in a predator-rich environment.

**Keywords:** Florida Keys; *Haemulidae*; macroinvertebrate; predator avoidance

## 1. Introduction

Predation arguably exerts a stronger controlling influence over the demography, density, and composition of benthic organisms than any other density-dependent process in aquatic and marine ecosystems [1–3]. As a result, many vulnerable organisms developed behavioral strategies that help to minimize encounter rates with higher order consumers. Perhaps the best known of these strategies is the preference of vulnerable organisms to occupy the shelter provided by structurally complex habitats to reduce the foraging efficiencies of visual predators [4], and references therein [5]. Other vulnerable organisms modify their feeding behavior and habitat occupancy patterns to reduce detection e.g., [6–8] during times when predators are nearby. In some cases, these types of risk-averse behavior can alter the strength of indirect interactions among organisms found in lower trophic levels e.g., [9].

Importantly, it should be noted that most of what we know about predator avoidance strategies in marine ecosystems is derived from studies conducted during daylight hours [10]. Far less is known about the importance of such strategies at night when mid-level consumers, themselves hiding in shelter during daylight hours, move out into nearby structurally simple habitats to feed [11–13]. Nocturnal invertebrate emergence is hypothesized to reduce such encounters with visual predators, in both benthic and pelagic

environments, in aquatic and marine ecosystems [14–17]. How widespread this potential predator avoidance strategy might be in marine ecosystems is less well-known [18,19], but it is reasonable to predict that this form of consumer avoidance is common and helps to maintain vulnerable lower order resilience given the primacy of predation in marine ecosystems [1,2].

Among the habitats that benthic macroinvertebrates may nocturnally emerge from are those located along the coral reef–seagrass interface, where predation is intense [11,17]. Permeable boundaries between such habitats, of greatly differing structural complexity, are frequently sites of intense predation [11]. During the day many vulnerable bottom-feeding fishes hide in the three-dimensional structure provided by coral reefs [20–23]. At night, mid-level consumers (e.g., grunts (*Haemulidae*)) leave the reef to feed in structurally simpler seagrass habitats and barren sand patches [20,24], but see [25]. It is this diurnal pattern of predator foraging that is hypothesized to trigger nocturnal prey emergence from seagrass habitats in widely separated regions of the world's ocean [17,26,27].

In a previously published study [17], we documented that nocturnal benthic macroin-vertebrate emergence from seagrasses do not vary with season, but unexpectedly do vary with distance from two geographically separated coral reefs located in the Florida Keys National Marine Sanctuary. This spatially variable pattern led us to hypothesize that the intensity of benthic emergence, performed to avoid detection and attack by actively feeding bottom-feeding predators, was controlled by differences in predator-foraging distance from the study reefs. Here, we report the findings of an additional laboratory study and of diurnal field surveys whose data were combined with emergence data used in a previously published study [17], to test the hypothesis that the variability of nocturnally active preda-tors explains the intensity of benthic macroinvertebrate emergence from seagrass habitats along the coral reef–seagrass interface [17]. To test this hypothesis, we: (1) documented the density and identity of mid-level reef-resident predators along the coral reef–seagrass interface during nighttime; (2) determined if spatial patterns of nocturnal emergence by benthic macroinvertebrates were positively correlated with the spatial abundance patterns of nocturnal predators; and (3) experimentally evaluated the impacts of the presence of nocturnal predators on macroinvertebrate emergence in laboratory microcosms.

## 2. Materials and Methods

### 2.1. Study Site Descriptions

The field component of this study was conducted at two widely separated sanctuary preservation areas (SPAs) in the upper (Key Largo Dry Rocks Reef) and middle (Newfound Harbor Reef; Figure 1) Florida Keys (hereafter referred to as "the Keys"). These are the same sites used in the initial study documenting the persistent occurrence of nocturnal macroinvertebrate emergence along the seagrass–coral reef interface in the Keys [17]. Briefly, Key Largo Dry Rocks SPA (latitude 25°07.59″, longitude 80°17.91″) contains a large contiguous seagrass bed, located on the leeward side of the reef. The second site, Newfound Harbor SPA (latitude 24°37.00″, longitude 81°23.86″) contains an aggregation of patch reefs and seagrasses meadows (for additional descriptions of these sites, see [28]).

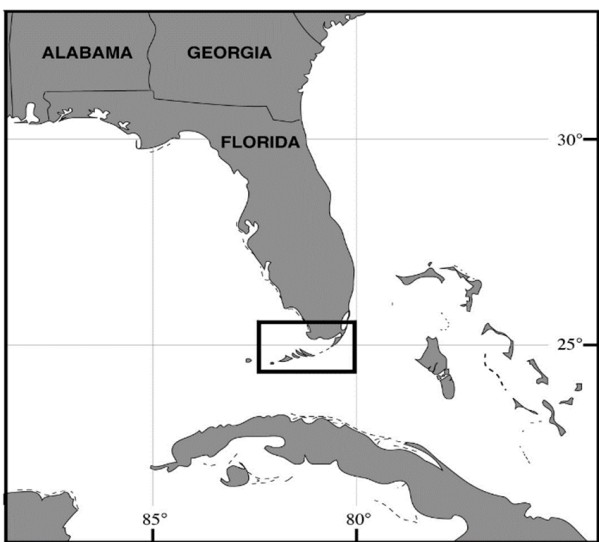

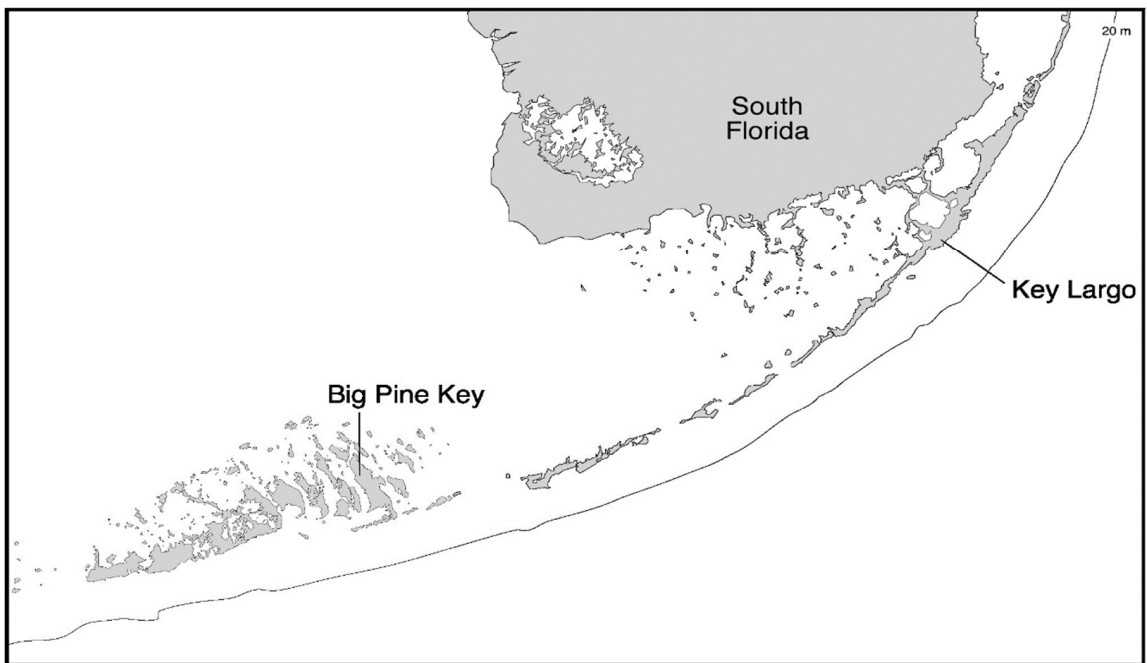

**Figure 1.** Study site locations.

### 2.2. Distributional Patterns of Nocturnally Active Predators

To determine if there is empirical support for the hypothesis that the presence of nocturnally active benthic predators triggers the emergence of macroinvertebrates from the benthos along the coral–seagrass interface, we documented the night-time density, composition, and distribution of predators at the two sites using replicate 20-m² visual belt transect censuses conducted directly next to the reefs and 30 m away [11].

At Dry Rocks, night-time belt transect surveys (*n* = four transects surveyed nightly for three consecutive days) were conducted concurrently with the deployment of emergence traps (*n* = 10 traps deployed nightly over the same three days) in May and August of 2002 and 2003 [17] for the results of the emergence study). At Newfound Harbor, belt transect surveys were replicated in May and September of 2002 and May of 2003 on the same nights the emergence traps were deployed.

### 2.3. Predator Impacts on Macroinvertebrate Emergence

To determine if nocturnally active mid-level predators could trigger macroinvertebrate emergence from the benthos, we conducted a second test under controlled laboratory conditions. We envisioned four alternative outcomes could occur in these experiments: (1) benthic macroinvertebrate emergence would not occur; (2) emergence would occur after sunset regardless of a predator's presence; (3) emergence would occur regardless of time of day when a predator was present; or (4) emergence would occur only after sunset when a predator was present during trials. To evaluate these alternatives, we conducted a factorially designed laboratory experiment consisting of four treatments: (1) a predator exclusion treatment conducted during daylight; (2) a predator inclusion treatment conducted during daylight; (3) a predator exclusion treatment conducted after sunset; and (4) a predator inclusion treatment conducted after sunset

Macroinvertebrates (polychaetes, amphipods, isopods, mysids, ostracods, and tanaids) used in these trials were collected using nocturnal plankton tows taken over seagrass beds adjacent to Newfound Harbor reef. These macroinvertebrates were the same numerically dominant phylogenetic groups collected in the emergence traps [17]. Upon return to the laboratory, macroinvertebrates were separated according to the taxonomic groups mentioned above, then allowed to acclimate to laboratory conditions for 24 h.

Experiments were conducted in replicate 0.038 L microcosms whose bottoms were covered with ~2 cm of azoic sand. Each microcosm was divided into two sections using 500 μm mesh cloth that allowed water movement throughout the tank yet kept the predator (when placed in one section of the tank) and the prey physically separated. Each trial began with the inoculation of 10 macroinvertebrates, randomly selected from one of the six previously mentioned taxonomic groups, into one randomly selected side of the microcosm. The macroinvertebrates were given 5 min to burrow into the sediments. One predator was placed in the compartment on the opposite side of the container in the predator inclusion treatments. Each treatment was replicated 10 times.

After a 10 min trial period, emergent macroinvertebrates were removed from the water column with a 500 υ mesh dip net. At the end of the trial, visual inspections of each tank were conducted, with the aid of a flashlight during night trials, to ensure that all the emergent organisms were collected. The netted macroinvertebrates were then counted using a dissecting microscope.

Based on the results of the belt transects, numerically abundant, blue-striped grunts (*Haemulon sciurus*) were selected to serve as predators in these laboratory experiments (see Belt Transect Results). Neither the macroinvertebrates, nor the blue-striped grunts, were reused in experiments more than two times. When predators were reused, they were not used in consecutive trials. Once trials were completed, all organisms were returned to local waters.

### 2.4. Statistical Analyses

#### 2.4.1. Field Surveys

Due to differences in the hydrography and the composition of coral reefs in the upper and middle Florida Keys [29,30], data collected at Dry Rocks and Newfound Harbor were analyzed separately. Emergence data used in these analyses came from the companion study that assessed the spatial and temporal variability of benthic macroinvertebrate emergence at our study sites [17].

Spatial and temporal patterns of fish abundance along the belt transects were compared using 2-factor analyses of variance (ANOVA)'s. The dependent variable was total fish density. Treatments included survey month and distance from the reef (near or far). These data satisfied the assumption of homogeneity of variances following $\log_{10}$ transformation. Levene's test for equality of variances was used in these, and all other, evaluations of the homogeneity of variance assumption for ANOVA. Total macroinvertebrate density in the emergence traps was analyzed as a function of month and distance from the reef (near or far). Data collected from Dry Rocks satisfied the assumption of homogeneity of variances,

while data collected from Newfound Harbor required square root transformation to satisfy this assumption [31].

Separate simple linear regressions of daily counts of total macroinvertebrate emergence on daily measurements of total fish density were performed for each site. Treatment effects in these and all other comparisons were significant when $p < 0.05$.

### 2.4.2. Laboratory Experiments

A 2-factor ANOVA was used to assess impacts of a predator's presence and time of day on macroinvertebrate emergence in the laboratory. The proportion of macroinvertebrates emerging into the water column was the dependent variable. Time of day (day or night) and predator presence/absence were the treatments. Data were arcsin square root transformed to satisfy the assumption of homogeneity of variance [31].

## 3. Results

### 3.1. Fish Density and Composition at Dry Rocks Reef

Grunts comprise >75% of all fishes recorded along the belt transects at Dry Rocks, with blue-striped (*Haemulon sciurus*), French (*H. flavolineatum*), unidentifiable immature grunts, and white grunts (*H. plumeri*) comprising ~35%, ~20%, ~18%, and <2% of all grunts counted, respectively. Other fish are present in smaller numbers, such as bar jacks (*Caranx ruber*) and unidentifiable silversides (Atherinidae), which represent 11% and 9.7% of all other fish counted, respectively. A 2-factor ANOVA conducted on $\log_{10}$ transformed fish densities counted along these belt transects failed to detect a significant interaction between survey month and distance from the reef ($F_{2, 30} = 1.78$, $p = 0.186$), or a significant effect of survey month on the number of fishes along the belt transects ($F_{2, 30} = 2.08$, $p = 0.142$), indicating that the spatial distribution patterns of nocturnally active fish remains similar throughout the course of the study. Fish density is significantly greater on belt transects placed furthest from the reef ($F_{1, 30} = 143.67$, $p < 0.001$) compared to those placed near the reef.

### 3.2. Patterns of Fish Density and Composition at Newfound Harbor

The composition of fish recorded on belt transects placed at Newfound Harbor is similar to that of Dry Rocks. Grunts are the most abundant fishes, accounting for over 80% of all individuals encountered during the nighttime belt transect surveys. Among the grunts, blue-striped grunts account for 42% of all observed fish species, French grunts for nearly 28% of all fishes counted, white grunts for over 3%, and some 8.5% smaller grunts were unidentified. The only other fish present in any substantial numbers are the silverside (accounting for ~12% of all fishes counted).

A 2-factor ANOVA on $\log_{(10)}$ transformed estimates of fish density along the belt transects detects a significant interaction between month and distance from the reef ($F_{2, 26} = 17.21$, $p < 0.001$). This interaction is the result of much lower fish counts recorded away from the reef in May 2003 than in May 2002 or September 2003, and greater numbers of fish being present near the reef during all three sampling periods.

### 3.3. The Relationship between Fish Density and Macroinvertebrate Emergence at Dry Rocks and Newfound Harbor Reefs

There is a highly significant positive linear relationship between macroinvertebrate emergence (i.e., the numbers of organisms collected in emergence traps) on fish density recorded at Dry Rocks Reef ($F_{1, 5} = 24.63$, $p = 0.007$). A very high proportion of the observed variation in macroinvertebrate numbers collected in the emergence traps is explained by variation in fish density ($r^2 = 0.86$; Figure 2a).

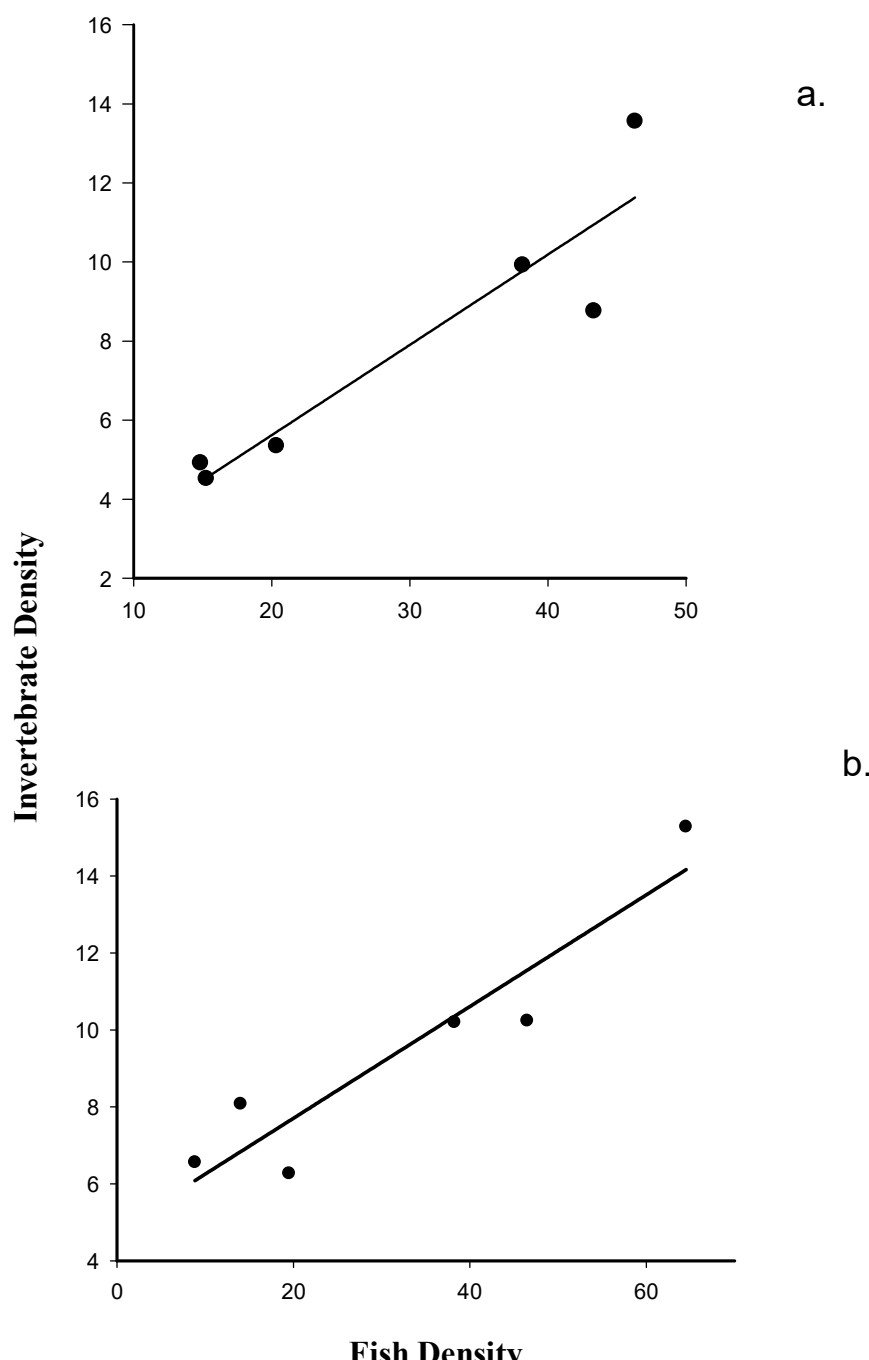

**Figure 2.** (**a**). Model II regression of benthic macroinvertebrate emergence on nocturnal fish counts at Dry Rocks Reef; and (**b**). Model II regression of benthic macroinvertebrate emergence on nocturnal fish counts at Newfound Harbor Reef.

The regression of transformed macroinvertebrate emergence on transformed fish density meets the test requirements for homogeneity of variance ($p$ = 0.060) tests. Similar to the results recorded at Dry Rocks, the results of this analysis detect a highly significant positive linear relationship between macroinvertebrate emergence and fish density at Newfound Harbor ($F_{1, 5}$ = 29.5, $p$ = 0.006). A very high proportion of the observed variation in emergence density is also explained by variation in fish density ($r^2$ = 0.89; Figure 2b) at this site.

### 3.4. Laboratory Experiments

Separate 2-factor ANOVAs conducted on untransformed data show that both time of day and a predator's presence (Table 1) have significant effects on the density of macroinvertebrates emerging into the water column. The presence of a predator triggers significantly more macroinvertebrates to leave the sediments in our experiments, regardless of time of day. That said, the emergence density of all six major macroinvertebrate groupings is significantly greater at night than during the day, and these differences are even greater in four of the six macroinvertebrate groups used in this experiment when a predator is present (Figure 3a–f). No significant 2-factor interactions are detected.

**Table 1.** Results of the analyses of variance conducted on the proportion of benthic invertebrates emerging from sediments in the presence/absence of a predator and time of day in the microcosm experiments. Results are considered significant when $p < 0.05$.

| Source | Mean Square | Degrees of Freedom | F | Sig. |
|---|---|---|---|---|
| **Ostracods** | | | | |
| Time | 45,630.55 | 1 | 513.77 | <0.0001 |
| Predator | 10,141.29 | 1 | 114.18 | <0.0001 |
| Time × predator | 318.06 | 1 | 3.58 | 0.065 |
| Error | 88.815 | 46 | | |
| **Polychaetes** | | | | |
| Time | 40,084.77 | 1 | 171.38 | <0.0001 |
| Predator | 4461.07 | 1 | 19.07 | <0.0001 |
| Time × predator | 140.79 | 1 | 0.602 | 0.442 |
| Error | 233.89 | 45 | | |
| **Amphipods** | | | | |
| Time | 31,482.12 | 1 | 449.6 | <0.0001 |
| Predator | 10,331.55 | 1 | 147.55 | <0.0001 |
| Time × predator | 103.87 | 1 | 1.483 | 0.229 |
| Error | 70.02 | 46 | | |
| **Isopods** | | | | |
| Time | 35,502.1 | 1 | 229.14 | <0.0001 |
| Predator | 6340.79 | 1 | 40.93 | <0.0001 |
| Time × predator | 445.22 | 1 | 2.874 | 0.097 |
| Error | 154.94 | 45 | | |
| **Mysids** | | | | |
| Time | 20,661.5 | 1 | 147.17 | <0.0001 |
| Predator | 598.26 | 1 | 4.26 | <0.022 |
| Time × predator | 163.62 | 1 | 1.165 | 0.322 |
| Error | 140.39 | 42 | | |
| **Tanaids** | | | | |
| Time | 51,055.82 | 1 | 317.43 | <0.0001 |
| Predator | 1365.16 | 1 | 8.49 | <0.006 |
| Time × predator | 29.55 | 1 | 0.184 | 0.67 |
| Error | 160.84 | 45 | | |

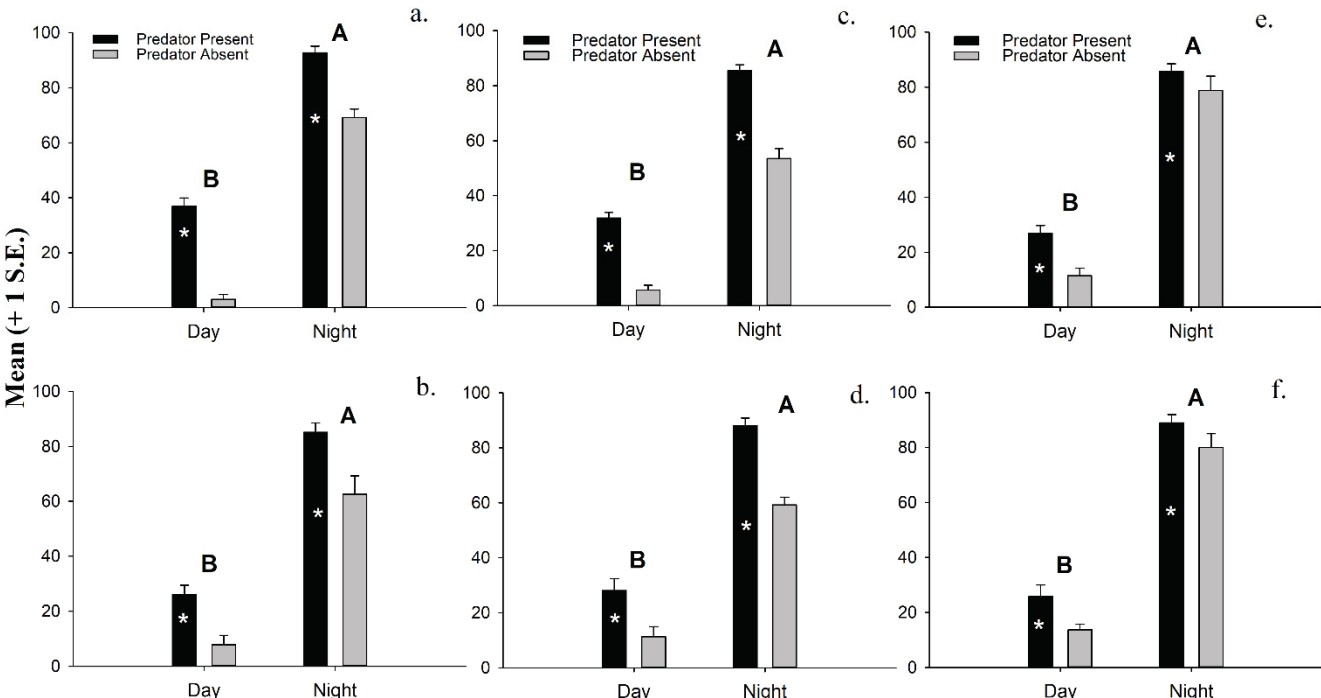

**Figure 3.** (**a**). Macroinvertebrate emergence density (mean ± 1 S.E.) of (**a**). ostracods, (**b**). polychaetes, (**c**). amphipods, (**d**). isopods, (**e**). mysids, and (**f**). tanaids in the laboratory experiments. * Indicates significant differences between treatments (*p* < 0.05).

## 4. Discussion

The results of this study show that the presence of bottom-feeding predators can trigger benthic macroinvertebrate emergence from the seagrasses that surround coral reefs. This pattern has previously been shown for zooplankton communities in freshwater and marine ecosystems, as well as for insects in aquatic benthic communities e.g., [4–6,32,33], and likely plays an important role in determining the diel transfer of energy across adjacent seagrasses and coral reefs.

In our previous study [17], we predicted that the presence of nocturnally active, bottom-feeding, reef-associated predators, those which migrate from coral reefs to feed at night, would trigger the emergence of seagrass-associated benthic macroinvertebrates into the overlying water column [15,17]. We also predicted that emergence intensity would be greatest in seagrasses closest to reefs, where we anticipated these predators would be most abundant, and lowest in seagrasses away from reefs where predator abundance was expected to be lower. The predictions were met at one site, Newfound Harbor, but not at the other, primarily because the distributional patterns of demersal predators were different.

In this study, we found that the presence of nocturnally active predators is positively related to prey emergence at both sites. In the laboratory experiments, the presence of these nocturnal predators also triggers emergence during daylight hours, but to a much lesser extent. This latter finding tends to confirm that the presence of a normally nocturnally active predator can play a role in triggering benthic emergence. Simple linear regressions of macroinvertebrates collected in the traps on fish densities explain >85% of variation in emergence intensity recorded at both sites. These highly significant results provide very strong support for our hypothesis that it is the nightly migration of bottom-feeding fishes that triggers the emergence of macroinvertebrates from seagrasses into the water column.

The results of the laboratory experiments strongly support the findings of the field studies, with nocturnal macroinvertebrate emergence from microcosm sediments being dramatically greater in the presence of a predator than in the absence of a predator, and more so at night in four of the six groups employed in the laboratory study. It should be noted that these results are strikingly similar to those found in studies of the nocturnal movement

patterns of aquatic insects in streams and rivers, as well as zooplankton in the pelagic reaches of lakes and the ocean [34]. In these studies, nocturnal migration patterns of small macroinvertebrates and aquatic insects are strongly linked to the presence of nocturnally active predators [32,35]. These findings, from studies conducted in widely separated study sites, argue strongly for the inclusion of nocturnal interactions in future predator–prey studies [10,23]. These results, when coupled with observation made elsewhere, suggest that seagrass biodiversity resilience may be reinforced by nocturnal prey avoidance strategies.

The results from this study have important implications for the management of tropical marine protected areas. Spatial subsidies of prey and nutrients, such as is provided by seagrass habitats, will likely sustain greater densities of higher order consumers in marine protected areas than would otherwise be possible if feeding was limited to the production of coral reefs cf. [36]. The results of the study reported on here identify a new, previously unconsidered, allochthonous pathway, driven by density dependent interactions, for the introduction of seagrass production into nearby unvegetated and coral reef habitats [11,37]. If currents carry these prey organisms over nearby reefs, as we expect, we hypothesize that predator-induced emigrations of seagrass-associated macroinvertebrates into the water column, and their subsequent redispersal, represents an as yet unrecognized form of cross-habitat energy exchange. This introduction of drifting macroinvertebrates, coupled with direct transfers of energy to the reef by nocturnally active bottom-feeding fishes and the grazing of seagrass leaves by parrotfishes, suggests that trophic links between seagrasses and coral reefs are substantial, and are critical for the successful management of marine protected areas [11,28,37].

Finally, most of what we know about the factors that determine the strength of predator–prey interactions along habitat boundaries comes from studies conducted during daylight hours, or at the end of a 24-h daytime deployment. Results from this study show that night-time predator–prey interactions in these habitats are also an important, and yet inadequately quantified, component of trophic interactions along the coral reef–seagrass boundary. This further suggests that adding data from nocturnal studies to marine protected area management plans may increase their likelihood of increasing predator biomass.

**Author Contributions:** Conceptualization, D.C.B. and J.F.V.; methodology, D.C.B. and J.F.V.; software, J.F.V.; validation, D.C.B. and J.F.V.; formal analysis, D.C.B. and J.F.V.; investigation, D.C.B. and J.F.V.; resources, D.C.B. and J.F.V.; data curation, D.C.B. and J.F.V.; writing—original draft preparation, D.C.B. and J.F.V.; writing—review and editing, J.F.V.; visualization, D.C.B. and J.F.V.; supervision, J.F.V.; project administration, DB.; funding acquisition, D.C.B. and J.F.V. All authors have read and agreed to the published version of the manuscript.

**Funding:** The development of these ideas and the actual conduct of this study were supported by grants from University of North Carolina at Wilmington National Undersea Research Center (UNCW #9537), The Nature Conservancy's Ecosystem Research Program, Marine Fisheries Initiative Program (MARFIN), PADI foundation, Project AWARE, and Sigma Xi.

**Institutional Review Board Statement:** Not applicable.

**Data Availability Statement:** All data are available upon request from the authors.

**Acknowledgments:** We express our appreciation to S. Miller and UNCW Captains Kendall Boykin and Otto Rutten for providing us with a historical background on fishing and the establishment of marine reserves in the Florida Keys National Marine Sanctuary. Additionally, we thank Richard Aronson, Neil Hammerschlag, Kenneth L. Heck, Brian Helmuth, Charles Martin, and Sean Powers, plus two anonymous reviewers, for constructive criticisms of this work. We also would like to thank the following individuals for their extensive help in the field: Geremea Fioravanti, Meg Goecker, Lindsey Kramer, Ryan Kroutil, and Bradley J. Peterson.

**Conflicts of Interest:** The funder organizations had no role in the design of the study; in the collection, analyses, or interpretation of data; in the writing of the manuscript, or in the decision to publish the results.

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
