# Peer review of "Predator-Induced Nocturnal Benthic Emergence: Field and Experimental Evidence for an Unknown Behavioral Escape Mechanism along the Coral Reef–Seagrass Interface"

_diversity, doi:10.3390/d14090762_

Round 1

Reviewer 1 Report

see attached file

Reviewer 2 Report

Major comment

The study provides contributions on predator-prey interactions, with evidence that mid-level predators can trigger the emergence of seagrass-associated benthic macroinvertebrates into the water column. The study contains a suitable sampling design coupling laboratory experiments and field surveys. While I think the article is interesting, and I agree with its main conclusion, I feel that it needs to consider that the nocturnal emergence of macroinvertebrates may be a strategy to avoid daytime predators. Also, I suggest some rearrange in the methods, a re-check in the figures cited in the text, and a minor organization of the results section. Please see below specific comments.  

Abstract

I suggest making clear for the reader that the manuscript deals with the emergence of macroinvertebrates from the benthos into the water column. “Nocturnal emergence” is also used for organisms leaving shelters to explore the benthos.

Line (L.) 17 – 20. What variability means here? Fish abundance? Please, clarify.

Introduction

The introduction is well-written, and no change is needed.  

Methods.

L. 101 – 104: Please, provide the total number of belt transects per month at Dry rocks and Newfound Harbor. Similarly, provide the number of emergence traps in each site and sampling occasion. I am afraid I didn’t understand – belt and emergence traps were placed concurrently only at Dry rocks?  I see that there is a previous study, but I believe that it is important giving more details on sampling in this this study.

L. 167 – 168. I suggest informing here that these tests were performed for each macroinvertebrate group separately.  It was not clear until I read the results.

Results.

L. 189-190. Assumption information can be better placed in the line 164 (statistical analysis).

L. 194 – I suggest indicating “Figure 2a” instead of only “Figure 2” in the parenthesis.

L. 194 – 196. I think this is better for the discussion section.

L. 187 – The section 3.2 is related to Dry Rocks, however, the Fig. 2 (Line 194 and L. 198) also shows the relationship between macroinvertebrate emergence and fish density for Newfound Harbor Reef (Fig. 2b). I am not sure if the solution is just to add Fig. 2a at Line 194 and replace Fig. 5 (L. 226) by Fig. 2b. My suggestion is to merge the section 3.2 and 3.4 into a single section, referring the Fig. 2a and Fig. 2b in the text.

L. 201 – The text of this section (3.3) named “Patterns of macroinvertebrate emergence intensity and fish density at Newfound Harbor” gives no information on macroinvertebrate intensity. All sentences (202 – 213) are related to fish composition and fish density estimates. Also, Fig 3. (cited in line 213) is about the laboratory experiments, but its text (Line 211 – 213) referred to fish counts.

L. 226 and L. 236. The number of figures is probably wrong, or some figures are missing. There are only 3 figures in the manuscript, but authors referred to figures 4 – 6. Figure 2 b relates to Newfound Harbor Reef but are placed in the section 3.2 (Dry Rocks).

Discussion

I agree that the results on this manuscript allowed authors to state that bottom-feeding mid-level predators can trigger macroinvertebrate emergence from the bottom to the water column. However, I think that it is important to consider in the discussion that the emergence into the water column can increase the predation risk by plankton feeders, some of them highly abundant at night in coral reefs. For instance, the nocturnal activity of Pempherids is associated with the abundance of the meroplanktonic crustaceans that emerge from the bottom into the water column after sunset.  More importantly, the nocturnal macroinvertebrate emergence may be a strategy to avoid predation by diurnal planktivorous fishes. A major factor that contributed to the nocturnal disposition of multiple species was probably the capacity to avoid daytime active predators. Are there planktivorous fish species in the studied seagrasses? How abundant are they? I feel that the discussion needs to consider that the nocturnal emergence of macroinvertebrates may be a strategy to avoid daytime predators.
